# Gene Expression Analysis and Validation of a Novel Biomarker Signature for Early-Stage Lung Adenocarcinoma

**DOI:** 10.3390/biom15060803

**Published:** 2025-05-31

**Authors:** Sanjan S. Sarang, Catherine M. Cahill, Jack T. Rogers

**Affiliations:** Neurochemistry Laboratory, Massachusetts General Hospital, Harvard Medical School, Boston, MA 02129, USA; ccahill@mgh.harvard.edu (C.M.C.); jackrogers@mgh.harvard.edu (J.T.R.)

**Keywords:** lung adenocarcinoma, differential gene expression, multiomics, protein–protein interactions, early-stage markers

## Abstract

Lung cancer is responsible for 2.21 million annual cancer cases and is the leading worldwide cause of cancer-related deaths. Specifically, lung adenocarcinoma (LUAD) is the most prevalent lung cancer subtype resulting from genetic causes; LUAD has a 15% patient survival rate due to it commonly being detected in its advanced stages. This study aimed to identify a novel biomarker signature of early-stage LUAD utilizing gene expression analysis of human lung tissue samples. Using 22 pairs of LUAD and matched normal lung microarrays, 229 differentially expressed genes were identified. These genes were networked for their protein–protein interactions, and 44 hub genes were determined from protein essentiality. Survival analysis of 478 LUAD patient samples identified four statistically significant candidates. These candidate genes’ expression profiles were validated from GTEx and TCGA (347 normal, 483 LUAD samples); immunohistochemistry validated the subsequent protein presence. Through intensive bioinformatic identification and multiple validations of the four-biomarker gene signature, AGER, MGP, and PECAM1 were identified as downregulated in LUAD; SLC2A1 was identified as upregulated in LUAD. These four biologically significant genes are involved in tumorigenesis and poor LUAD prognosis, meriting their use as a clinical biomarker signature and therapeutic targets for early-stage LUAD.

## 1. Introduction

Cancers are a modern threat to society and are classified by the uncontrolled growth and spread of abnormal cells throughout the body that can result in death, either with or without treatment [1]. Lung cancer, responsible for 2.21 million annual cases, is the leading worldwide cause of cancer-related deaths, with prognosis tightly linked to stage at diagnosis [2,3]. Lung cancers can be classified into two main subgroups: non-small-cell lung cancer (NSCLC), responsible for 85% of all lung cancer cases, and small-cell lung cancer, responsible for 15% of all lung cancer cases [2]. Specifically, lung adenocarcinoma (LUAD) is responsible for ten percent of all NSCLC cases in patients with no smoking history, suggesting that the incidence of LUAD is related to genetic variations and susceptibility [4]. In normal lung cells, proto-oncogenes are regulators of cell growth. However, modifications can occur in these genes, forming oncogenes and leading to carcinogenesis. These oncogenes can form through changes such as chromosomal translocation (e.g., chromosomal translocation of the *MYC* oncogene in Burkitt’s lymphoma), point mutation (e.g., point mutation at codon 12 of the RAS oncogene), and gene amplification (e.g., amplification of *c-MYC* in neuroblastoma) [5]. A second pivotal gene type that is involved in the pathogenesis of cancer is tumor suppressor genes. In normal cells, these genes regulate cell growth and block cancer; mutations that reduce the presence or activation of these genes lead to cancer. For example, *P53* is a tumor suppressor gene that is a tetramer and, when mutated, can lead to significant effects as the tetramer will always contain at least one mutant *P53* protein [5]. In LUAD, genetic abnormalities occur in normal alveolar and bronchial epithelial cells, promoting precancerous lesions such as atypical adenomatous hyperplasia (AAH). Epidermal growth factor receptor or anaplastic lymphoma kinase mutations within these lesions lead to elevated cell proliferation that allows these cells to survive and accumulate mutations [6]. Further mutations then cause these AAH lesions to progress into adenocarcinoma in situ (AIS), a non-invasive stage characterized by cellular atypia and architectural distortion [7]. With the AIS lesions acquiring more genetic instability, the lesions become invasive adenocarcinoma, penetrating the basement membrane and spreading to surrounding tissues [8].

The identification of markers in LUAD is critical as it can allow for diagnosing LUAD in its earlier stages and may enable timely intervention [9]. Low-dose CT (LDCT) screening can improve early detection and reduce mortality by approximately 20% in high-risk smokers. However, it suffers from frequent false positives and overdiagnosis [3,10]. These issues underscore a specific need for biomarkers to refine the screening and diagnosis of LUAD. Furthermore, utilizing a biomarker signature, a composite of multiple biomarkers can enhance diagnostic accuracy by integrating numerous molecular insights and minimizing variability [11]. Previous biomarker identification studies have been performed using bioinformatic analysis of gene data or laboratory tests and have discovered many biomarkers for various diseases [12,13,14,15,16]. Despite this, very few clinically recognized biomarkers for early-stage LUAD [12,13] are currently available. Identifying biomarkers for early-stage LUAD diagnosis is essential, as LUAD is a prominent cancer usually detected in its advanced stages [13,17]. At these advanced stages, there is a 15% survival rate. Early detection of LUAD through an accurate biomarker could allow for treatments to be conducted on a less advanced tumor, leading to an improved patient outcome [12,13,17].

This research aimed to identify a predictive biomarker signature of unique cancer-related genes that can be utilized for the early-stage diagnosis of LUAD to reduce disease mortality. In this investigation, we identified a novel biomarker signature to aid in predicting poor LUAD prognosis via early-stage samples.

Currently, many clinically recognized biomarkers of LUAD, such as carcinoembryonic antigen and CYFRA21-1, have limited sensitivity and specificity considering an application in early-stage disease [18]. Our research identified a biomarker signature comprising *AGER*, *MGP*, *PECAM1*, and *SLC2A1*, which uniquely integrates various relevant pathways such as cell adhesion, angiogenesis, and metabolic reprogramming, which improves the overall specificity and diagnostic confidence as opposed to single-marker approaches [19]. The signature utilizes the novel markers *AGER* and *MGP*; the downregulation of these markers has rarely been explored in LUAD [20,21]. In summary, the utilization of a biomarker signature with relevant markers increases diagnostic accuracy beyond single-marker approaches [10].

## 2. Materials and Methods

### 2.1. Data Acquisition

The transcriptomics dataset GSE32863 was obtained from the Gene Expression Omnibus (GEO), and the dataset was analyzed using the GEO2R tool. The dataset comprises gene expression profiles from 58 primary LUAD tumor tissues and 58 matched adjacent normal lung tissues [13]. The analysis was focused on analyzing 22 tumors classified as stages I-II and the 22 associated matched normal samples. All microarray data were derived from a high-density oligonucleotide platform that covers approximately 21,000 genes [13]. Each sample was assigned labels via the define feature, and the grouping was verified before proceeding with the analysis.

### 2.2. Differential Gene Expression Analysis

To accurately identify differentially expressed genes (DEGs), the Benjamini–Hochberg adjustment method was applied to the GEO2R analysis tool. Additionally, a highly stringent adjusted *p*-value (adj. *p*) cutoff of adj. *p* ≤ 0.01 was utilized to ensure statistical significance. Furthermore, a Log_2_ Fold Change (Log_2_FC) cutoff of | Log_2_FC | ≥ 2 was used to determine up- and downregulated DEGs [22]. Utilizing R version 4.3.2 (R Foundation for Statistical Computing, Vienna, Austria) data quality and normalization were verified by evaluating median-centered expression distributions for all selected samples in numerous comparative boxplots. The combination of cutoffs and validation of median-centered ensured that only the most significant and biologically relevant changes were selected for further downstream analysis.

### 2.3. Protein–Protein Interaction Networks and Hub Gene Identification

Once all genes were categorized as upregulated or downregulated, the entire set was uploaded into the Search Tool for the Retrieval of Interacting Genes/Proteins (STRING v11.5; https://string-db.org, ELIXIR Core Data Resource, EMBL, Heidelberg, Germany) to construct a protein–protein interaction (PPI) network. The network was then exported into Cytoscape v3.26.0 (The Cytoscape Consortium, San Diego, CA, USA; https://cytoscape.org), where the cytoHubba plugin v0.1 (developed by Institute of Systems Biology, National Tsing Hua University, Hsinchu, Taiwan) was used to identify hub genes. Using the two most robust centrality algorithms, Maximal Clique Centrality (MCC) and Density of Maximum Neighborhood Component (DMNC), both algorithms computed the shortest path to identify hub genes. In constructing the protein–protein interaction network, interactions with a confidence score of ≥0.4 were considered significant participants in the network [23]. Using the two topological algorithms, MCC and DMNC, nodes were ranked by centrality; a combination of both computations was used to determine the final hub gene set for further analysis. This approach considered overlapping high-centrality nodes from both methods, which elevated the confidence that the hub genes were pivotal in the LUAD network.

### 2.4. Kaplan–Meier Survival Analysis

Hub genes were exported into the Gene Expression Profiling Interactive Analysis 2 (GEPIA-2, v2.0; http://gepia2.cancer-pku.cn, Peking University, Beijing, China) platform to generate Kaplan–Meier survival curves, which visualize the correlation between gene expression levels with the overall survival (OS) of LUAD patients. The LUAD patient data were imported from The Cancer Genome Atlas (TCGA) and Genotype-Tissue Expression (GTEx) database. Genes that were observed to have significant survival correlation were retained, thus highlighting their prognostic potential in earlier-stage LUAD. Genes with log-rank *p*-values < 0.05 in the Kaplan–Meier analysis were deemed to be associated with overall survival. The default median expression cutoff in GEPIA-2 allowed for the high and low expression groups to be dichotomized [24]. The genes that met the criterion were retained as candidates for prognostic potential, allowing the subsequent analysis to focus on the most robust survival correlations.

### 2.5. Transcriptomic and Proteomic Validation

After OS analysis, the smaller set of key genes were validated for their overall differential expression using the “Expression DIY” tool in GEPIA-2, which accessed TCGA and GTEx for LUAD and healthy tissue expression data. Due to the high specificity of antigen–antibody binding reactions, immunohistochemistry (IHC) stains from the Human Protein Atlas (HPA) (https://www.proteinatlas.org/, Science for Life Laboratory, Uppsala University, Uppsala, Sweden) were used to validate the presence of proteins encoded by the biomarker genes. For the immunohistochemical validation, lung tissue staining was retrieved for each available candidate gene from the HPA [25]. The HPA provided results from three independent normal lung tissue subjects and five to six independent LUAD tumor tissue subjects for each gene, enabling a comparison between protein presence in normal and cancerous tissues. The staining level and intensity strength were reported.

## 3. Results

### 3.1. Identification of DEGs

The GSE32863 expression profile was downloaded from GEO and included LUAD tumor and normal lung samples for 22 unique patients, which assayed 21,044 distinct genes per sample as described by Selamat et al. [13]. The boxplot, shown in Figure 1a, was also analyzed and displayed a precise median centering. GEO2R generated a volcano plot of every gene demonstrated in Figure 1b and predicted the Log_2_FC and *p*-value for each DEG shown in Table A1. Using |Log_2_FC| ≥ 2 as a differential expression cutoff, 229 DEGs were identified, of which 80 were upregulated in LUAD and 149 were downregulated in LUAD (Table 1).

### 3.2. PPI Analysis of DEGs

Using the STRING database, 229 DEGs were used to construct a PPI network to show interacting proteins, resulting in 227 interacting nodes and 445 edges (Figure A1). The network was exported to Cytoscape for analysis, where cytoHubba was used to screen for hub genes. These are represented by their high connectivity and interactions with other genes in LUAD, as shown by the proteins or nodes with more interactions or edges (Figure 2). CytoHubba’s most robust algorithms, MCC and DMNC, produced two separate PPIs of the hub genes shown in Figure 2. From the two PPIs with some overlapping identifications, 44 total hub genes were identified (Table 2).

### 3.3. Survival Analysis of Hub Genes

Using GEPIA-2, the correlation between hub gene expression and LUAD patient survival could be made using 478 patients accessed from TCGA and GTEx. OS analysis indicated that the underexpression of *AGER* (*p* = 0.002), *MGP* (*p* = 0.00093), and *PECAM1* (*p* = 0.0035) correlated with lower survival. Conversely, OS indicated that the overexpression of *SLC2A1* (*p* = 2.4 × 10^−5^) leads to lower survival (Figure 3). From the analysis of all 44 hub genes, *AGER*, *MGP*, *PECAM1*, and *SLC2A1* can be significantly correlated with the prognosis of LUAD, meaning these genes are candidate biomarkers for the early detection of LUAD.

### 3.4. Transcriptomic Verification of Key Genes Using LUAD Samples from TCGA and GTEx

GEPIA-2 was used to generate differential expression boxplots to validate the candidate biomarker genes for their mRNA differential expression between LUAD and normal lung tissues. GEPIA-2 accessed LUAD and normal lung gene expression data from TCGA and GTEx databases, and the data were used as a test dataset; 483 patients were accessed for tumor tissue gene expression data, and 347 patients were accessed for normal tissue gene expression data. GEPIA-2 produced differential expression boxplots and evaluated for statistical significance to show that *AGER*, *MGP*, and *PECAM1* were downregulated in LUAD compared to normal tissues (Figure 4). GEPIA-2 verified that *SLC2A1* was upregulated in LUAD compared to normal tissues (Figure 4). This verification revealed that *AGER*, *MGP*, and *PECAM1* are significantly downregulated in LUAD and that *SLC2A1* is significantly upregulated in LUAD.

### 3.5. Proteomic Validation of Key Genes Using Immunohistochemistry

Using the HPA, AGER, MGP, PECAM1, and SLC2A1, normal lung and LUAD tumor IHC staining was accessed. The protein presence was represented by the amount of IHC staining within the normal human lung tissue and LUAD tumor tissue sample. Intensity and level of staining was observed across the biopsy samples. Figure 5 is representative of the overall findings. The IHC showed that AGER and PECAM1 have high intensity and strong staining in all normal lung samples (3/3 subjects), whereas in LUAD tumor tissues, these proteins were absent (0/5–6 showed any positive staining). SLC2A1 had no staining in the normal lung samples (0/3) but had strong expression in the 4/6 LUAD tumor cases. Matching normal lung tissue and LUAD tumor tissue IHC staining for MGP presence was unavailable. According to the extensive bioinformatic analysis, MGP is a highly accurate candidate for its function as a gene biomarker. Overall, AGER, PECAM1, and SLC2A1 were validated for their expression in normal and LUAD tumor IHC samples, showing that the candidate gene biomarkers can function as biologically significant biomarkers to aid in the detection of early-stage LUAD. The robustness of these tissue biomarkers is reinforced by transcriptomic analyses, underscoring their prognostic potential.

## 4. Discussion

This study aimed to identify and validate a novel biomarker signature for early-stage LUAD. We successfully determined a biomarker signature composed of *AGER*, *MGP*, *PECAM1*, and *SLC2A1* that can be utilized to accurately diagnose early-stage LUAD. Using GEO2R, 229 DEGs were identified and networked by the STRING database, and cytoHubba was used to screen 44 hub genes using the MCC and DMNC algorithms. Overall survival (OS) analysis was conducted to verify the correlation of each hub gene’s expression with LUAD prognosis using a TCGA and GTEx test dataset of 478 patient survival data points. *AGER*, *MGP*, *PECAM1*, and *SLC2A1* were the key genes significantly correlated with poorer LUAD prognosis. GEPIA-2 was used to validate the genes’ expression in LUAD by accessing 483 LUAD tumor tissue expression patients and 347 normal lung tissue patients from TCGA and GTEx to act as a test dataset. Lastly, HPA was used to validate the downstream protein expression of each gene using IHC stains of both normal and LUAD tumor tissue. The cumulative approach of identifying genetic biomarkers using GEO, TCGA, GTEx, and HPA allows for the usage of a biologically significant biomarker signature for early-stage LUAD.

There have been several recent studies that identify early diagnostic markers or gene signatures for LUAD that each highlight varying aspects of tumor biology. Li et al. in 2023 used a proteomic profile to propose a panel of secreted proteins that included midkine, WFDC2, and CXCL14 as candidates for early-stage markers for LUAD [9]. Chen et al. performed an integrated analysis of various transcriptomic datasets that elucidated cell cycle-related genes, e.g., *ASPM* and *CCNB2*; these genes were upregulated and associated with a poorer LUAD prognosis [26]. These studies highlight that early-stage LUAD is characterized by diverse molecular changes and provides valuable context for interpreting our biomarker signature.

Superior to other panels, the four-gene signature identified in this study does not overlap with the above candidates, highlighting its novelty [18]. This signature is critically unique because it emphasizes the loss of certain tumor-suppressive factors alongside a known oncogenic driver. *AGER*, *MGP*, and *PECAM1* are significantly downregulated. Downregulating *AGER* and *PECAM1* is highly notable because of their involvement in maintaining normal alveolar cell adhesion and vascular integrity in the lung. The loss of these two genes could potentially facilitate the disorganized growth and angiogenesis that characterizes incipient tumors [20,27]. The upregulation of *SLC2A1* in the signature shows a development in the Warburg effect that can even occur in earlier tumor stages, which is consistent with alternate reports that link *SLC2A1* to aggressive lung cancer behavior [28]. The integration of gene expression, patient survival data, and proteomic validation showed a comprehensive validation of these markers’ clinical relevance. Compared to the existing literature that relied on mRNA data alone, these results demonstrate their significance through rigorous cross-validation.

We list our top four biomarkers as follows: *AGER*, *MGP*, *PECAM1* (downregulated), and *SLC2A1* (upregulated). Advanced Glycosylation End-Product Specific Receptor (AGER) is a transmembrane multi-ligand receptor in the immunoglobulin superfamily. *AGER* is involved in almost every cell type; *AGER* acts as an adhesion molecule in lung epithelium, creating contact between alveolar type I cells and their substrate [20]. This study showed *AGER* is underexpressed in LUAD compared with normal lung tissues; underexpression was correlated with a poorer cancer prognosis (Figure 3). IHC analysis also revealed that AGER had no staining in the LUAD tumor and had high staining in the normal lung, validating the downregulation of the gene in LUAD (Figure 5). *AGER* downregulation has been linked to the loss of cell differentiation and epithelial structure organization simultaneously with oncogenic transformation [20]. Furthermore, existing studies link *AGER* to a poorer prognosis of LUAD, highlighting its biomarker potential [29].

Matrix Gla Protein (MGP) is a protein in the extracellular matrix near vascular tissues. *MGP* is a calcification inhibitor that maintains normal vascular function [30]. This study showed that *MGP* is underexpressed in LUAD compared to normal lung tissues; underexpression was correlated with a poorer cancer prognosis (Figure 3). *MGP* downregulation has been linked to elevated bone morphogenic protein signaling, which is connected to arterial–venous malformations and elevated angiogenesis. *MGP* has also been shown to become upregulated in later-stage cancers to increase tumor stabilization and perfusion [30,31].

Platelet And Endothelial Cell Adhesion Molecule 1 (PECAM1) is a transmembrane protein, part of the immunoglobulin superfamily [32,33]. *PECAM1* connects adjacent endothelial cells and regulates inflammation, leukocyte migration, and vascular responses during sepsis [32]. This study showed *PECAM1* is underexpressed in LUAD compared to normal lung tissues, and the underexpression was linked to poorer cancer prognosis (Figure 3). IHC analysis also revealed that PECAM1 had no staining in the LUAD tumor and had high staining in the normal lung, validating the downregulation of the gene in LUAD (Figure 5). *PECAM1* downregulation has been shown to mediate the secretion of metallopeptidase inhibitor 1, a protein linked to tumor cell proliferation [27].

Solute carrier family 2-facilitated glucose transporter member 1 (SLC2A1) is a glucose transporter highly concentrated in tissue endothelium and epithelium. *SLC2A1* is involved in glucose uptake and allows for aerobic glycolysis [34]. This study showed that *SLC2A1* is overexpressed in LUAD compared to normal lung tissues, and this overexpression was linked to a poorer cancer prognosis (Figure 3). IHC analysis also revealed that SLC2A1 had high staining in the LUAD tumor and had no staining in the normal lung, validating the upregulation of the gene in LUAD (Figure 5). *SLC2A1* overexpression has been linked to transporting more glucose across the cell membrane and allowing for the continuation of aerobic glycolysis and the cell cycle, aiding tumor cell proliferation [34].

Circulating cell-free DNA (cfDNA) is being increasingly utilized in cancer diagnosis, including LUAD. The biomarker signature identified in this study, more specifically AGER and MGP, has unique methylation patterns in tumor versus normal tissues [13,21]. These differences in methylation are detectable in cfDNA and can provide a minimally invasive diagnostic potential to complement imaging techniques such as LDCT screening effectively [35]. Future research may investigate the early-detection performance of these biomarkers in patient plasma cfDNA samples.

Therapeutic strategies targeting the signature could aim to restore the downregulated tumor suppressors or potentially inhibit the upregulated oncogenic pathways in LUAD. Low-dose DNA methyltransferase inhibitors, such as 5-azacytidine, can reverse the gene silencing in lung tumors; analogs of retinoic acid have been shown to halt proliferation mediated by *AGER* and *MGP* [36,37]. The metabolic vulnerability highlighted by the upregulation of *SLC2A1* can be targeted through inhibitors such as WZB117, which impair glucose uptake and could reduce LUAD cell proliferation [38]. The downregulation of *PECAM1* is indicative of an enhanced angiogenesis, which can be mitigated by the multi-kinase inhibitor nintedanib [39]. Targeted small-molecule antagonists of *AGER* show promise in disrupting tumor-promoting *AGER*-mediated signaling [40]. Taken together, these strategies for modulating significant mediators of LUAD prognosis can support the translational potential of this biomarker signature.

The presented biomarker signature shows significant potential as a diagnostic tool in clinical practice. A combined immunohistochemical assay that incorporates AGER, MGP, PECAM1, and SLC2A1 could significantly aid pathology-based distinction between malignant and benign lesions in lung biopsy. Moreover, loss of AGER and PECAM1, coupled with an elevated SLC2A1 in staining, could identify these malignancies effectively in earlier stages of LUAD [29,41].

Additionally, blood-based detection assays that identify altered methylation in AGER and MGP, as well as circulating soluble proteins such as soluble AGER (sRAGE) and MGP, offer minimally invasive options. sRAGE shows potential in allowing for differentiation between LUAD and healthy individuals due to its consistently observed reduction in malignancy [42]. To reduce the prevalence of false positives, these minimally invasive tests could be used as a complement to clinical imaging.

## 5. Conclusions

This study identified and validated a biomarker signature of *AGER*, *MGP*, *PECAM1*, and *SCL2A1* as key genes linked to tumorigenesis and poorer LUAD prognosis. The correlation of these genes to LUAD suggests their clinical usage as biomarkers in detecting early-stage LUAD and are possible targets for therapy. In vivo studies can be used to determine the specific role of these markers in tumor growth, invasion, and response to therapy to further validate these markers.

## Figures and Tables

**Figure 1 biomolecules-15-00803-f001:**
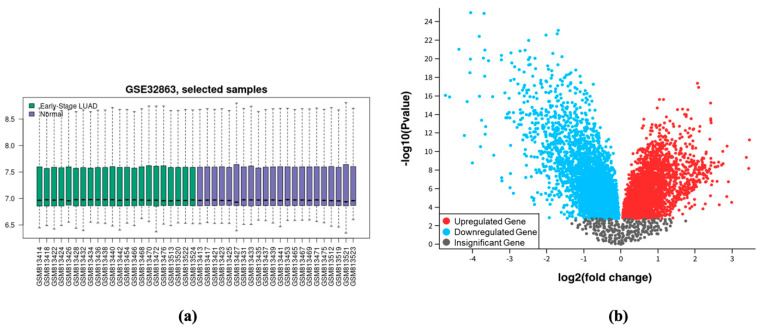
Boxplot of sample expression and volcano plot of DEGs from GSE32863. (**a**) Boxplot of selected samples: median centering shows the statistical correction to be accurate and the samples ready for further analysis. (**b**) Volcano plot of DEGs: each dot represents a gene regulation and significance prediction.

**Figure 2 biomolecules-15-00803-f002:**
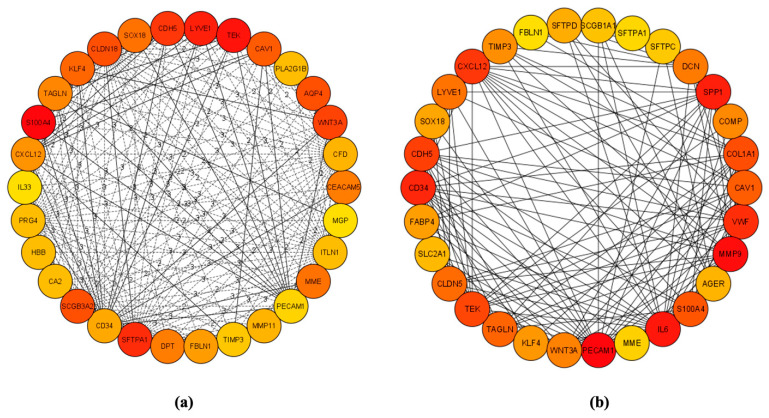
PPI network of hub gene interconnectedness from Cytoscape. All edges (lines) represent protein interactions between specific nodes (proteins). (**a**) DMNC algorithm hub gene predictions; (**b**) MCC algorithm hub gene predictions.

**Figure 3 biomolecules-15-00803-f003:**
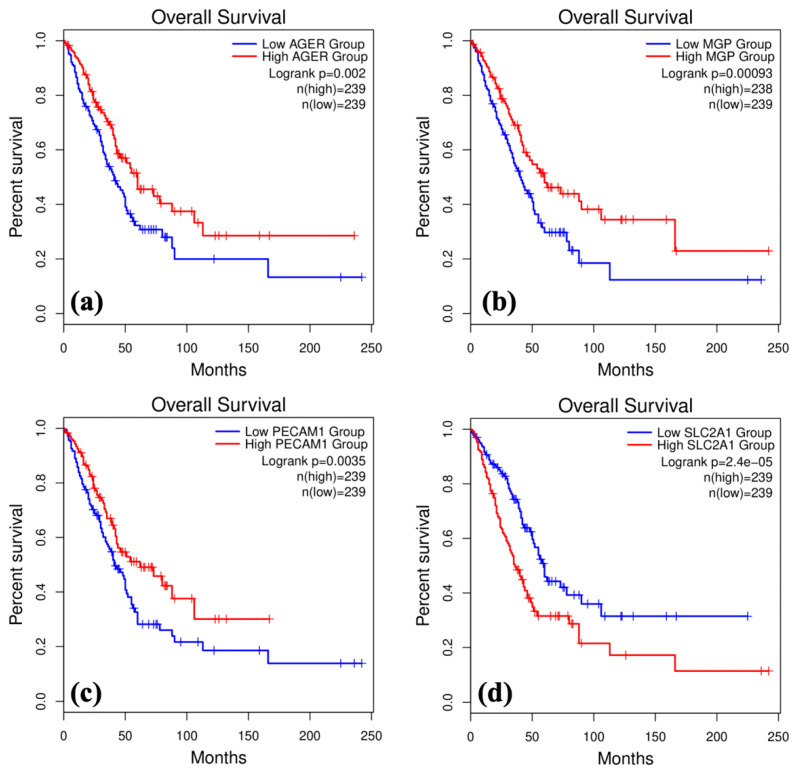
Kaplan–Meier survival analysis of candidate markers. (**a**) *AGER* survival analysis. (**b**) *MGP* survival analysis. (**c**) *PECAM1* survival analysis. (**d**) *SLC2A1* survival analysis.

**Figure 4 biomolecules-15-00803-f004:**
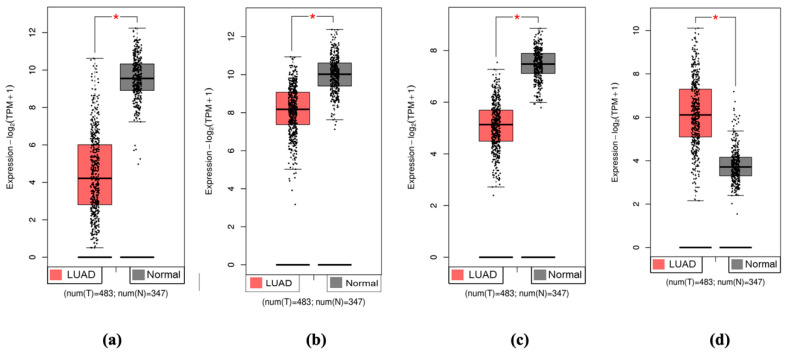
Validation of gene regulation with TCGA and GTEx databases. (**a**) *AGER* downregulation validation. (**b**) *MGP* downregulation validation. (**c**) *PECAM1* downregulation validation. (**d**) *SLC2A1* upregulation validation. * *p* < 0.01.

**Figure 5 biomolecules-15-00803-f005:**
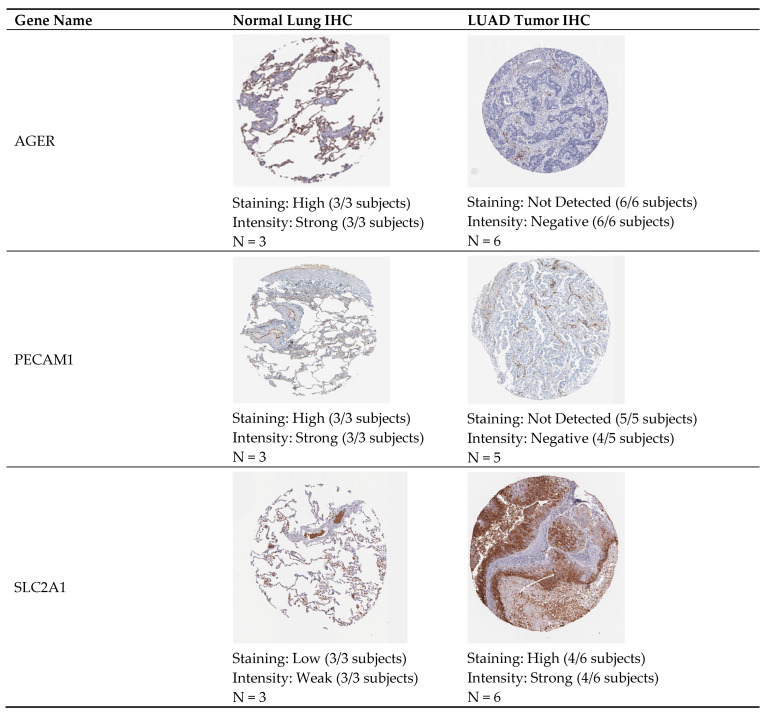
Representative normal lung and tumor cell IHC staining from HPA.

**Table 1 biomolecules-15-00803-t001:** Set of upregulated and downregulated genes from GEO2R.

Gene Regulation	DEG Names
Upregulated *	*GCNT3*, *CEACAM5*, *SPP1*, *CST1*, *SPINK1*, *CRABP2*, *MMP11*, *XRCC2*, *COMP*, *SEMA3E*, *FKBP14*, *CHRNA5*, *SHROOM4*, *SFN*, *NLRP8*, *LMOD3*, *LAD1*, *PROM2*, *GSDMB*, *EEF1A2*, *SERINC2*, *TDP1*, *MBTD1*, *ITCH-IT1*, *FCGBP*, *FMC1*, *DTWD2*, *IL17RD*, *HYPK*, *USP49*, *MMP9*, *SEZ6L2*, *APOPT1*, *ZNF682*, *HSD17B7*, *ZNF483*, *ZNF69*, *COL1A1*, *EXO5*, *MAGT1*, *HNRNPU*, *GPR1*, *TM4SF4*, *SLC2A1*, *ZNF14*, *METTL21A*, *ALPP*, *DENR*, *ZNF394*, *BLZF1*, *SPDEF*, *DMC1*, *LRRFIP1*, *MIGA1*, *MDK*, *AOC4P*, *CEACAM1*, *SSTR2*, *TMEM17*, *SLC35E1*, *ZNF577*, *SGPP2*, *CAPN8*, *CEP19*, *DDX51*, *MCMDC*, *PYCR1*, *YRDC*, *WDR74*, *OCIAD1*, *CCBE1*, *N4BP*, *PTGR2*, *DUSP19*, *TOP2A*, *EID2B*, *TRIM13*, *TNFSF1*, *POFUT1*, *PODXL2*
Downregulated *	*ENPP2*, *C11orf96*, *GYPC*, *WNT3A*, *MS4A7*, *CALCRL*, *PLPP3*, *SOSTDC1*, *SFTPD*, *JAM2*, *RASL12*, *TAGLN*, *CD34*, *GRK5*, *STOM*, *ABI3BP*, *CD52*, *BCHE*, *SMAD6*, *HYAL1*, *SLPI*, *TPSAB1*, *FBLN1*, *DPT*, *FCN1*, *KLF4*, *SOCS2*, *PPP1R14A*, *SDCBP*, *PTGDS*, *FPR1*, *VSIG4*, *ADAMTS1*, *CLIC5*, *CYYR1*, *VWF*, *STX11*, *RASIP1*, *CA2*, *LAMP3*, *FEZ1*, *MGP*, *RAMP3*, *PDK4*, *PLA2G1B*, *HOXA5*, *EPAS1*, *PI16*, *S100A4*, *CXCL12*, *GAS1*, *GPC3*, *MAL*, *CLEC14A*, *CES1*, *FABP5*, *MME*, *IL33*, *ANOS1*, *C7*, *ITLN1*, *DNASE1L3*, *SCGB3A2*, *PECAM1*, *FHL1*, *ITM2A*, *EDNRB*, *FAM110D*, *ID3*, *MAMDC2*, *S100A8*, *SVEP1*, *C9orf24*, *SRPX*, *ACTG2*, *AQP4*, *TSC22D1*, *CRYAB*, *MMRN1*, *CD300LG*, *PCOLCE2*, *TSPAN7*, *COX7A1*, *ABCA8*, *CDH5*, *PRG4*, *SOX18*, *CD93*, *PGM5*, *SRGN*, *CYP4B1*, *MARCO*, *ADH1B*, *WIF1*, *SDPR*, *EFEMP1*, *HIGD1B*, *CPB2*, *C2orf40*, *TNNC1*, *CD36*, *RGCC*, *CAV2*, *LDB2*, *TIMP3*, *CLDN5*, *SEPP1*, *SPARCL1*, *CPA3*, *TEK*, *PEBP4*, *CFD*, *FMO2*, *IL6*, *SPOCK2*, *CRTAC1*, *SFTPC*, *AGER*, *DCN*, *LYVE1*, *GPIHBP1*, *MT1M*, *CLDN18*, *PGC*, *TMEM100*, *MFAP4*, *ADIRF*, *ACKR1*, *INMT*, *GNG11*, *FOSB*, *SCGB1A1*, *CCL14*, *TCF21*, *PLAC9*, *GKN2*, *CAV1*, *SFTPA1*, *HBA1*, *ADH1A*, *CLEC3B*, *FABP4*, *HBA2*, *FCN3*, *FAM107A*, *ITLN2*, *MCEMP1*, *CA4*, *HBB*

* Adj. *p* < 0.01.

**Table 2 biomolecules-15-00803-t002:** Hub genes from MCC and DMNC algorithms based on PPI network.

Hub Gene Name
*VWF*, *DCN*, *FBLN1*, *MME*, *S100A*, *SCGB1A1*, *COMP*, *CXCL12*, *TEK*, *TAGLN*, *AGER*, *SLC2A1*, *IL6*, *SFTPC*, *CLDN5*, *SPP1*, *SFTPD*, *CDH5*, *PECAM1*, *CD34*, *COL1A1*, *CAV1*, *LYVE*, *FABP4*, *WNT3A*, *TIMP3*, *KLF4*, *MMP9*, *SFTPA1*, *SOX18*, *MGP*, *DPT*, *ITLN1*, *IL33*, *SCGB3A2*, *CFD*, *CA2*, *PRG4*, *MMP11*, *CEACAM5*, *AQP4*, *PLA2G1B*, *HBB*, *CLDN*

## Data Availability

The datasets analyzed during the current study are available in the Gene Expression Omnibus under accession number GSE32863 (https://www.ncbi.nlm.nih.gov/geo/query/acc.cgi?acc=GSE32863, accessed on 22 September 2024) and are published by Selamat et al. (DOI: 10.1101/gr.132662.111).

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
