# Peer review of "Gene Expression Analysis and Validation of a Novel Biomarker Signature for Early-Stage Lung Adenocarcinoma"

_biomolecules, 2025, doi:10.3390/biom15060803_

Round 1
Reviewer 1 Report
Comments and Suggestions for Authors
The manuscript by Sarang et al. reports the results of gene expression analysis and validation of a novel biomarker signature for early stage lung adenocarcinoma. Although the manuscript reports some novel data, it raises questions that should be addressed prior to publication of the results obtained:
- What are the advantages of the newly identified biomarkers over existing ones? For example, the introduction of the manuscript lacks a description of the field, while the discussion lacks comparisons of the data obtained with existing methods.
- How do the authors envisage the use of their findings in clinical practice?
- 3.5. Proteomic validation of key genes by immunohistochemistry: How many samples have been analyzed? Are all experimental samples really positive with "strong intensity"?
- The Materials and Methods section does not provide sufficient information to reproduce most of the experiments reported.
Author Response
Comments 1: The manuscript by Sarang et al. reports the results of gene expression analysis and validation of a novel biomarker signature for early stage lung adenocarcinoma. Although the manuscript reports some novel data, it raises questions that should be addressed prior to publication of the results obtained:
- What are the advantages of the newly identified biomarkers over existing ones? For example, the introduction of the manuscript lacks a description of the field, while the discussion lacks comparisons of the data obtained with existing methods.

- How do the authors envisage the use of their findings in clinical practice?

- 3.5. Proteomic validation of key genes by immunohistochemistry: How many samples have been analyzed? Are all experimental samples really positive with "strong intensity"?

- The Materials and Methods section does not provide sufficient information to reproduce most of the experiments reported.
Response 1:
Dear Reviewer 1,
Thank you very much for your detailed review of our manuscript. We have revised the manuscript accordingly, making the following changes. We have included a section in the introduction that provides a thorough review of the limitations of existing LUAD biomarkers and expands on the description of the field (Lines 33-34; 59-63; 77-86). In the discussion, a comparison of the findings with existing methods was included (Lines 249-271). A section was also added which envisages the utilization of this novel biomarker panel in clinical practice through a combined immunohistochemical assay and potentially blood-based detection assays to complement existing low-dose CT screening techniques (Lines 310-316; 329-340). In Section 3.5, we have included additional information regarding the number of subjects analyzed from the Human Protein Atlas (Lines 210-217). Figure 5 was also updated to describe the level and intensity of staining for each subject in AGER, PECAM1, and SLC2A1. (Lines 221-232). In the Materials and Methods section, additional details were provided to address the reproducibility of the experiment (Lines 90-95, 102-106, 114-120, 128-132, 139-144).
Reviewer 2 Report
Comments and Suggestions for Authors
This manuscript is well constructed and of considerable interest. Graphical features are well presented. In the discussions section, I wonder if you could compare/contrast results with other reported gene expression patterns from other authors? Might there be application in analysis of peripheral cfDNA in lung cancer? I appreciated the potential of practical application suggested (IHC).
Author Response
Comments 2: This manuscript is well constructed and of considerable interest. Graphical features are well presented. In the discussions section, I wonder if you could compare/contrast results with other reported gene expression patterns from other authors? Might there be application in analysis of peripheral cfDNA in lung cancer? I appreciated the potential of practical application suggested (IHC).
Response 2:
Dear Reviewer 2,
Thank you for your detailed review of our manuscript. We have revised the manuscript accordingly, making the following changes. We have added a section in the discussion detailing the results of this biomarker study to other recently reported gene expression patterns (Lines 249-271). Furthermore, we have included a paragraph on the potential application of peripheral cfDNA in lung cancer identification from patient plasma to complement the possible use of the proposed biomarker panel and low-dose CT screening (Lines 310-316). We have also added a section on the potential use of blood-based detection assays via altered methylation patterns in AGER and MGP; soluble AGER is a promising route to alternative LUAD identification techniques (Lines 335-340).
Round 2
Reviewer 1 Report
Comments and Suggestions for Authors
the manuscript has been improved